# Spatial and Temporal Variations in Fish Assemblage in Feitsui Reservoir, in Northern Taiwan, from 2006–2020

**Chen Yi-Ron [1], Hou Wen-Shang [2], Huang Chen-Kang [1] and Chou Chu-Yang [1],***

[1] Department of Biomechatronics Engineering, National Taiwan University, Taipei 10617, Taiwan; r89622042@ntu.edu.tw (C.Y.-R.); ckhuang94530@ntu.edu.tw (H.C.-K.)

[2] Department of Bioenvironmental Systems Engineering, National Taiwan University, Taipei 10617, Taiwan; houws@ntu.edu.tw

* Correspondence: choucy@ntu.edu.tw; Tel.: +886-2-3366-5333

**Abstract:** Reservoirs are artificial ecosystems that modify the hydrological and environmental components nearby. The long-term monitoring of fish assemblages in reservoirs may provide key information to understand how these artificial ecosystems allow fish population fluctuations and identify proper conservation strategies. A sentinel-site approach method assessed changes in fish assemblages in the Feitsui Reservoir (1024 ha) over 14 years, including the periods 2006–2008, 2010–2011, 2016, 2018, and 2020. Fish assemblages, sampled using gill nets, were composed of 38 species (12 families and 8 orders) with Cyprinidae as the dominant family; the fish fauna were dominated by species of the family Cyprinidae (71%) and Cichlidae (20.3%). Principal component analysis and nonmetric multidimensional scaling categorized the assemblages into two groups (cold and warm seasons), and we identified three parameters that were significantly correlated with the season ($p < 0.05$): SD (R = −0.04), Chl-a (R = 0.01), and algal count (R = 0.19); the algal count was higher in the warm season than in the cold season. The fish assemblage in the cold and warm seasons contained no exclusive taxa, contributing to the dissimilarity between the groups. The fish assemblage for the years before and after 2010 indicated partial overlap between the two groups; *S. macrops* contributed greatly to the separation of the two periods (>10% each; SIMPER) and *O. mossambicus* was dominant in assemblage structures after 2010. Our findings show the importance of long-term fish monitoring for the investigation of the effects of nonnative fish species on native fish species composition.

**Keywords:** fish assemblages; long-term monitoring; persistence and stability; Feitsui Reservoir; temporal distributions



## 1. Introduction

The degree of spatial and temporal change in fish assemblage patterns may depend on the magnitude of environmental perturbations. Environmental variation has been identified as one of the major drivers of fish diversity patterns [1]. River damming has altered the fish biodiversity of many rivers, leading to considerable consequences concerning the structure and functioning of those natural ecosystems [2,3]. Fish assemblage structures have been assessed in many impounded reservoirs, with pronounced variations in assemblages being observed in different reservoirs [4–6]. Long-term fish assemblages studies have suggested that fish groups may transition to new states that eventually stabilize over several decades [7–10], whereas other processes (e.g., species invasions or dam operation) may cause variations in community structures [11]. Because of this, long-term monitoring is indispensable; it allows for an understanding of temporal changes in biodiversity. The long-term monitoring of species assemblage can provide valuable information on the main biotic forces that affect the structure of ecological communities and can thus enable the detection and assessment of the anthropogenic mechanisms underlying these processes [12–15].

Habitat fragmentation and flow regulation, which are generally considered separately, are widely recognized as the two most severe effects of dam construction [16–18]. Dams

are frequently configured constructed in a row on a single river or basin and thus form a reservoir waterfall [16]. Generally, four reservoir habitats can be observed in a cascading reservoir: the natural riverine stretches adjacent to reservoirs, the riverine zone, the transitional zone, and the lacustrine region [16,19,20]. Reservoirs are intermediate environments between rivers and lakes. Reservoirs are sometimes referred to as hybrid systems, involving complex interactions, and, therefore, more variable patterns [19]. Dam impoundment in the reservoir creates habitat environmental heterogeneity, with a longitudinal gradient stream forming in the upstream–downstream direction, which leads to significant differences in the physical and chemical characteristics in the riverine zone upstream of the reservoir and the lacustrine region in the reservoir [21]. Studies have demonstrated that changes in each physical and chemical feature directly influence the structure of fish assemblages; variations in characteristics, such as water level, flow velocity, and residence time, may drastically alter fish accumulation dynamics across the spatial scale, changing from longitudinal (e.g., rivers) to vertical (e.g., lakes) and vice versa [19,22–25]. Because of these habitat changes, reservoir areas are key research areas for determining short- and long-term changes in fish assemblage structures[5,21,26,27].

Herein, we present a long-term study of fish assemblage in a reservoir in northern Taiwan (similar to many reservoirs in the region) to assess whether this reservoir also contains the spatial gradients in fish assemblage attributes and structures that have been observed in larger reservoirs. Many lotic fish species, following the periodic changes in the water levels of reservoirs, move between the riverine and transitional region, whereas carnivorous and planktivorous fishes, which have adapted to lentic habitats, tend to occupy the lacustrine regions of the reservoirs [20,28]. In this study, we additionally quantified the long-term effects of the presence of nonnative fish species on native fish assemblages in reservoirs that have been studied at our sentinel locations based on data obtained through intensive year-round gillnet sampling.

The aim of this study was to investigate changes in fish assemblage structure in the Feitsui Reservoir, Taiwan, for 14 years post impoundment, using a sentinel-locations approach. We analyzed differences in fish assemblage throughout years to identify whether the long-term variation associated with environmental conditions was evident in the fish populations by (1) quantifying the extent and dynamics of spatial changes, and changes in the correlation between water quality and fish, and by (2) identifying changes in fish assemblage structures due to the influence of exotic fish species.

## 2. Materials and Methods

The Feitsui Reservoir is the second largest reservoir in Taiwan, at $0.046 \times 10^{10}$ m$^3$, with a surface area of 303 km$^2$. All samples in this study were collected in the Feitsui Reservoir from the same nine fixed stations (Figure 1). Fish congregations were sampled at nine locations along the river (range of approximately 23 km), overlaying the entire area affected by the impoundment (i.e., the fluvial, transition, and lacustrine region of the reservoir). All sites were located upstream from the dam. Location sites 1 and 2 were 1 and 4 km distant from the dam, respectively (lacustrine region). Sites 3, 4, 5, 6, and 7 were 6, 8, 10.5, 12.5, and 14 km distant from the dam, respectively (transition region). Sites 8 and 9 were located 17.3 and 20 km upstream of the dam, respectively (fluvial region).

### 2.1. Environmental Parameters

To characterize the natural conditions in the reservoirs and distinguish the various sites with respect to their water quality monitoring station, several parameters related to water chemistry and physics were obtained for all sites. Data for the reservoir and several water parameters were provided by the Taipei Feitsui Reservoir Administration (sampled at nine sites) and included Secchi disk depth (SD), water temperature (WT), turbidity, total dissolved solids (TDS), suspended solids (SS), pH, total phosphorus (TP), dissolved oxygen (DO), electrical conductivity (EC), concentration of chlorophyll-a (Chl-a), and algal count.

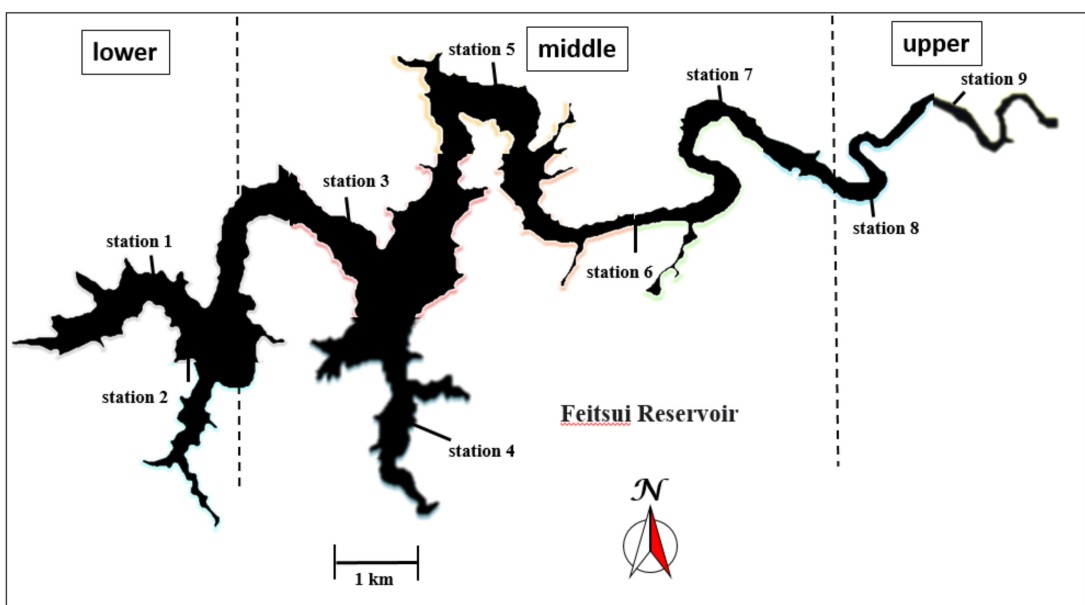

**Figure 1.** Location of the nine fixed sample stations in the Feitsui Reservoir (black lines).

### 2.2. Fish Sampling

Fish were sampled with gillnets at least 4–6 times to year during five periods: 2006–2008, 2010–2011, 2016, 2018, and 2020. We used the collected data to inspect patterns of long-term change in the fish fauna structure for Feitsui Reservoir. Our gillnets could not collect samples of the small-bodied fishes that are abundant in the reservoir. Therefore, our analyses were restricted to large-bodied classes and larger-sized species (i.e., large fish >100 mm total length). For fish sampling, the nets used were the same for the whole sampling, gillnets (mesh size: 4.5 to 20 cm between opposite knots) were deployed with lengths of 20–60 m and heights of 1.5–4.5 m at each reservoir. Two nets (one surface and one lower) were set at each site in the morning and were retrieved on the second day (average set time of 20 h). Different fishing gear was deployed along the vertical regions and cross sections of the river based on habitat type because of the differences in size and species selectivity of fishing gear. Fishing gear was positioned at different layers of the water (the surface, middle, and deep zones and the bed) and other locations (the littoral and open water zones) in the river cross sections. The bottom nets were set with the lead line directly on the bottom, and surface nets were set with the top line approximately 0.5 m below the surface. All fish samples were identified to the species, sorted, and counted following the Latin-Chinese Dictionary of Fishes Names [29]. For each fish, the total length (in cm) was measured using a tape ruler (up to 150 cm) and the total weight (in kg) was measured using an electronic balance (up to 50 kg).

### 2.3. Data Analysis

PCA (principal component analysis) is a method for common extracting data and reducing dimensionality. PCA is often used as a preprocessing step for subsequent analyses. In PCA, variables in the data set are grouped into a smaller set of influential variables through linear combinations. The original data can then be mapped onto new data vectors spanned by the principal new uncorrelated (orthogonal) variables, called principal components (PCs); the PCs can be used to successfully extract relevant information from the data. A few PCs with large variance explanation rates can inform the main characteristics of the raw data via a PCA ordination diagram. This two PCs with the largest variance explanation rates are selected from the sequence diagram as the coordinate axes, which enables the distribution of the feature, correlations, and distances between the samples to be simulated and analyzed. This allows for the description of entire data sets, which leads to data reduction with a minimal loss of information [30–33]. The per month WT

and the algal count at water quality monitoring stations on the Feitsui Reservoir from 2006 to 2020 were used to calculate the ten aforementioned indicator eigenvalues. The above-mentioned indicator eigenvalue for the PCA sequence diagram was used to determine water quality changes during the cold and warm seasons. The ordination diagram of PCA was drawn using PRIMER v6 software (Plymouth Marine Laboratory, Plymouth, UK). We considered a fish assemblage sampled in a given sampling year and performed nonmetric multidimensional scaling (MDS) ordination. Stress coefficients were counted as critical values to distance the goodness-of-fit of measure of MDS with respect to data and distances [34]. The differences in fish assemblages were calculated using an analysis of similarity (ANOSIM). A similarity percentage (SIMPER) was analyzed to identify the species most responsible for the differences in species year-groups [34]. The multivariate analyses were performed with the PRIMER v6 software package, which included MDS, ANOSIM, and SIMPER modules [35].

## 3. Results

### 3.1. Seasonal Environmental Variability

The data used in the PCA were collected over 14 y (from 2006 to 2020) and comprised 118 groups of measured values ($118 \times 10$) at the Feitsui Reservoir. According to the results of the PCA analysis, we selected the following variables as related to fish assemblage variation: SD (m), WT ($°C$), turbidity (NTU), pH, DO (mg $L^{-1}$), TDS (mg $L^{-1}$), EC ($\mu$S $cm^{-1}$), TP ($\mu$g $L^{-1}$), Chl-a ($\mu$g $L^{-1}$), and algal count (cell $mL^{-1}$). Our findings revealed a mean SD of $4.09 \pm 1.19$ m, WT of $24.13 \pm 4.75$ $°C$ at the surface, turbidity of $2.18 \pm 3.36$, pH of $7.52 \pm 0.56$ at the surface, and DO of $7.65 \pm 0.79$ mg $L^{-1}$ at the surface. EC was always low, with a mean of $66.81 \pm 7.52$ $\mu$S $cm^{-1}$. The results also revealed a mean TP of $13.20 \pm 7.52$ $\mu$g $L^{-1}$, Chl-a of $3.81 \pm 2.99$ $\mu$g $L^{-1}$, and algal count of $26,257 \pm 32,378$ cell $mL^{-1}$. Several of the parameters exhibited slight trends over time. SD, WT, pH, DO, Chl-a, and algal count changed with the cold and warm seasons. WT, pH, Chl-a, and algal count tended to increase in the warm season and decrease in the cold season. SD tended to rise with algal count, whereas DO decreased with increased WT. Turbidity and TP followed no notable trends.

From 2006 to 2020, WT and SD parameters were collected from the study area of nine stations. A mean WT of below or above 24 $°C$ was considered to indicate the cold or warm seasons, respectively. Notably, the results indicated that the cold and warm seasons influenced changes in the water quality of the Feitsui Reservoir. In addition to WT, we identified three parameters that were significantly correlated with the season ($p < 0.05$): SD (R = $-0.04$), Chl-a (R = 0.01), and algal count (R = 0.19). These parameters may be major sources of the changes in the water quality during the different seasons.

According to the PCA results, out of the 10 main components, only two PCs with eigenvalues higher than 0.27 were selected for multiple linear regression analysis. These selected PCs explained 60% of the total variation of variables in the PCA (Table 1). The component loadings from the PCA for the PCs are presented in Table 2. In Table 2, the bold loading indicate the highest correlations between variables and corresponding components. For example, turbidity, TP, and Chl-a, which demonstrated the highest correlation with PC2, were evaluated as a group, and algal count, which demonstrated the highest correlation with PC1, was independently assessed. The PC1 of the PCA (Table 1) explained 64.5% of data variability and correlated primarily with the algal count variable (0.973). The PC2 explained 80.6% of data variability and was associated with the Turbidity ($-0.426$), TP ($-0.487$), and Chl-a ($-0.668$). The indicators of the ordination diagram reflected the WT characteristics of the cold and warm seasons (Figure 2). In Figure 2, the blue indicators represent warm season WT characteristics, and the green indicators represent those of the cold season. The water quality changed along the PC1 axis; the indicators more closely associated with the PC1 axis were the increases or decreases in algal count. The algal count was higher in the warm season than in the cold, indicating that changes in algal count in the study area were more prominent when the WT was higher. The PC2 axis represents the

difference in the degree of eutrophication in the reservoir's upper, middle, and lower zones. The degree of eutrophication increased as the sample was drawn from further upstream.

**Table 1.** Descriptive statistics of selected PCs.

|  | PC1 | PC2 | PC3 | PC4 | PC5 |
|---|---|---|---|---|---|
| Eigenvalue | 1.12 | 0.279 | 0.146 | 0.115 | $2.99 \times 10^{-2}$ |
| Total variance (%) | 64.5 | 16.1 | 8.4 | 6.6 | 1.7 |
| Cumulative variance proportion (%) | 64.5 | 80.6 | 89.0 | 95.6 | 97.4 |

**Table 2.** Results of principal component analysis.

| | Loading of Variables | | | | |
|---|---|---|---|---|---|
| **Variables** | **PC1** | **PC2** | **PC3** | **PC4** | **PC5** |
| SD | −0.076 | 0.285 | −0.200 | 0.251 | −0.261 |
| WT | 0.102 | −0.062 | 0.045 | −0.007 | 0.837 |
| Turbidity | 0.061 | −0.426 | 0.421 | −0.673 | −0.193 |
| pH | 0.019 | −0.012 | −0.010 | −0.011 | 0.163 |
| DO | 0.007 | 0.052 | −0.010 | −0.016 | −0.277 |
| TDS | 0.025 | −0.022 | −0.014 | 0.027 | 0.102 |
| EC | −0.004 | −0.058 | −0.050 | 0.000 | 0.244 |
| TP | 0.093 | −0.487 | 0.509 | 0.694 | −0.091 |
| Chl-a | 0.154 | −0.668 | −0.720 | 0.027 | −0.072 |
| Algal count | 0.973 | 0.208 | 0.019 | −0.008 | −0.079 |

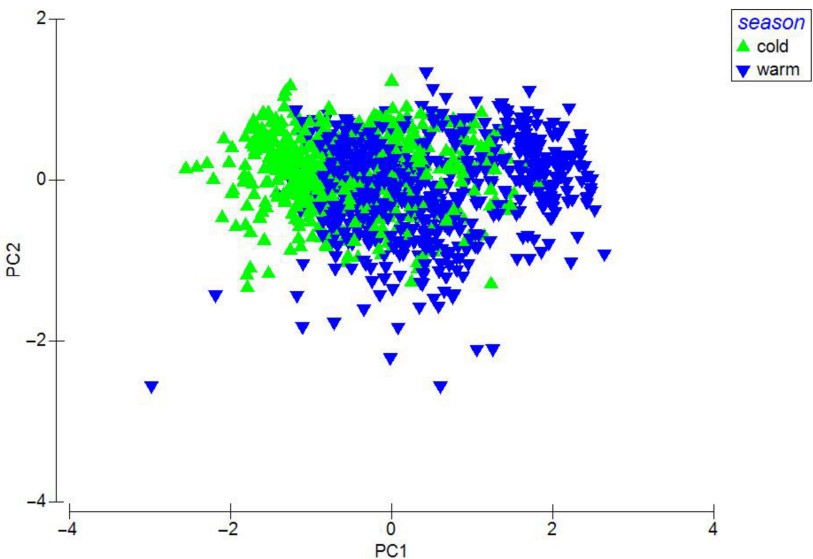

**Figure 2.** PCA axes for cold and warm seasons water temperature matrix (water temperature below 24 °C indicates cold season; water temperature above 24 °C indicates warm season).

*3.2. Fish Species Composition*

From 2006 to 2020, we gathered a total of 7247 specimens, comprising 38 species of 12 families and 8 orders. The fish communities were dominated by species belonging to Cyprinidae (71%) and Cichlidae (20.3%); numerically abundant species included *Sinibrama macrops* (12.6%), *Hemiculter leucisculus* (11.4%), *Cyprinus carpio* (11%), and *Chanodichthys erythropterus* (10.8%). An interannual variation in fish species between 19 and 29 (mean = 21.4, SD = 4.3) was recorded. The between-years correlation matrix of the fish species composition obtained from the pairwise test is presented in Table 3. Generally, the reservoir fish species compositions between years had relatively high and positive correlations for all 14 year based on the results of the pairwise test listed in Table 3. The

species composition in 2014 differed significantly from that in the other years; the species composition in 2016 was also significantly different from that in 2008, 2010, 2011, and 2014. The species composition in 2010 was significantly different from that in 2006, 2007, and 2020. In addition, the species composition in 2011 was significantly different from that in 2008 and 2020. SIMPER analysis demonstrated that nine species contributed to 10% of the dissimilarity. The species that contributed to the dissimilarity in 2010, 2011, 2014, 2016, 2018, and 2020 were *Hypophthalmichthys nobilis*, *Carassius cuvieri*, *S. macrops*, *C. carpio*, and *H. leucisculus*; *Distoechodon tumirostris*; *S. macrops*, *H. leucisculus*, *C. erythropterus*, and *Mozambique tilapia*; *M. tilapia* and *C. cuvieri*; *D. tumirostris*, *M. tilapia*, and *Parachromis managuensis*; and *S. macrops*, *P. managuensis*, and *M. tilapia*, respectively (Table 4).

**Table 3.** Between-years correlation matrix of fish species composition obtained from pairwise test.

| | | | | Pairwise Test | | | | |
|---|---|---|---|---|---|---|---|---|
| *p*-Value | 2006 | 2007 | 2008 | 2010 | 2011 | 2014 | 2016 | 2018 |
| 2007 | 0.537 | | | | | | | |
| 2008 | 0.476 | 0.8 | | | | | | |
| 2010 | 0.017 | 0.048 | 0.11 | | | | | |
| 2011 | 0.127 | 0.133 | 0.029 | 0.19 | | | | |
| 2014 | 0.019 | 0.004 | 0.029 | 0.002 | 0.005 | | | |
| 2016 | 0.167 | 0.305 | 0.029 | 0.024 | 0.029 | 0.019 | | |
| 2018 | 0.024 | 0.005 | 0.029 | 0.005 | 0.029 | 0.005 | 0.029 | |
| 2020 | 0.071 | 0.062 | 0.029 | 0.005 | 0.029 | 0.010 | 0.057 | 0.057 |

**Table 4.** SIMPER analysis on the fish species contributing to dissimilarity between species composition between years.

| Species | Contribution to Dissimilarity (%) | | | | | | | |
|---|---|---|---|---|---|---|---|---|
| | G 2006 vs. G 2010 | G 2007 vs. G 2010 | G 2008 vs. G 2011 | G 2006 vs. G 2014 | G 2011 vs. G 2014 | G 2010 vs. G 2016 | G 2011 vs. G 2018 | G 2011 vs. G 2020 |
| *Carassius cuvieri* | | 10.34 | | | | 10.16 | | |
| *Chanodichthys erythropterus* | | | | 12.68 | | | | |
| *Cyprinus carpio* | | 10.26 | | | | | | |
| *Distoechodon tumirostris* | | | 10.92 | | | | 11.52 | |
| *Hemiculter leucisculus* | 11.23 | | | 12.76 | | | | |
| *Hypophthalmichthys nobilis* | | 13.18 | | | | | | |
| *Mozambique tilapia* | | | | | 10.22 | 10.83 | 13.68 | 10.25 |
| *Parachromis managuensis* | | | | | | | 16.47 | 16.62 |
| *Sinibrama macrops* | 12.67 | | | 13.1 | | | | 17.88 |

### 3.3. Long-Term Variation in the Fish Assemblage

MDS indicated differences in the fish assemblage in three data sets (one for each period): 2008, 2011, and 2014 (Figure 3a). However, the MDS plot of the fish assemblage for the years before and after 2010 indicated a partial overlap between the two groups (Figure 3b). The cold and warm seasons contained no exclusive taxa, contributing to the dissimilarity between the groups (Figure 3c). A stress value of 0.19 indicated that the two-dimensional plot reasonably represented the multidimensional distances among data. ANOSIM verified the differences in fish assemblage structures between periods (Global R = 0.088, *p* < 0.05; Table 5). Fourteen species, *S. macrops*, *Culter erythropterus*, *Hemibarbus labeo*, *Ctenopharyngodon idella*, *C. cuvieri*, *Oxyeleotris marmorata*, *Aristichthys nobilis*, *D. tumirostris*, *H. leucisculus*, *Oreochromis* sp., *Carassius auratus auratus*, *Cyprinus carpio carpio*, *Silurus asotus*, and *Hypostomus* sp., were present in each year. These 14 taxa constitute 93% of total species abundance during our study. *S. macrops* contributed greatly to the separation of the two periods (>10% each; SIMPER).

**Table 5.** Cold and warm season similarity analysis (ANOSIM) for pre- and post 2010 years. ($p < 0.05$).

| Feitsui Reservoir | Cold and Warm Season | Pre and Post 2010 Period | Years |
| --- | --- | --- | --- |
| Global R | 0.088 | 0.168 | 0.369 |
| *p*-value | 0.100 | 0.001 | 0.001 |

Through SIMPER analysis, we were able to parse the overall responses of species populations that significantly contributed to changes in assemblage abundance between years. We then applied SIMPER analysis to identify species differences in the shift in assemblages between the years pre- and post 2010. These results indicated that five species contributed to 53.94% of the dissimilarity in the years pre- and post 2010 (Table 6) and that nine species decreased in abundance.

**Table 6.** SIMPER analysis of fish species contributing to similarity between species composition during pre- and post-2010 years.

| Groups Before & After | | |
| --- | --- | --- |
| **Average Dissimilarity = 53.94** | | |
| **Species** | **Contrib (%)** | **Cum. (%)** |
| *Sinibrama macrops* | 10.8 | 10.8 |
| *Oreochromis* sp. | 8.62 | 19.42 |
| *Culter erythropterus* | 8.07 | 27.49 |
| *Carassius auratus auratus* | 7.56 | 35.05 |
| *Hemiculter leucisculus* | 7.29 | 42.34 |

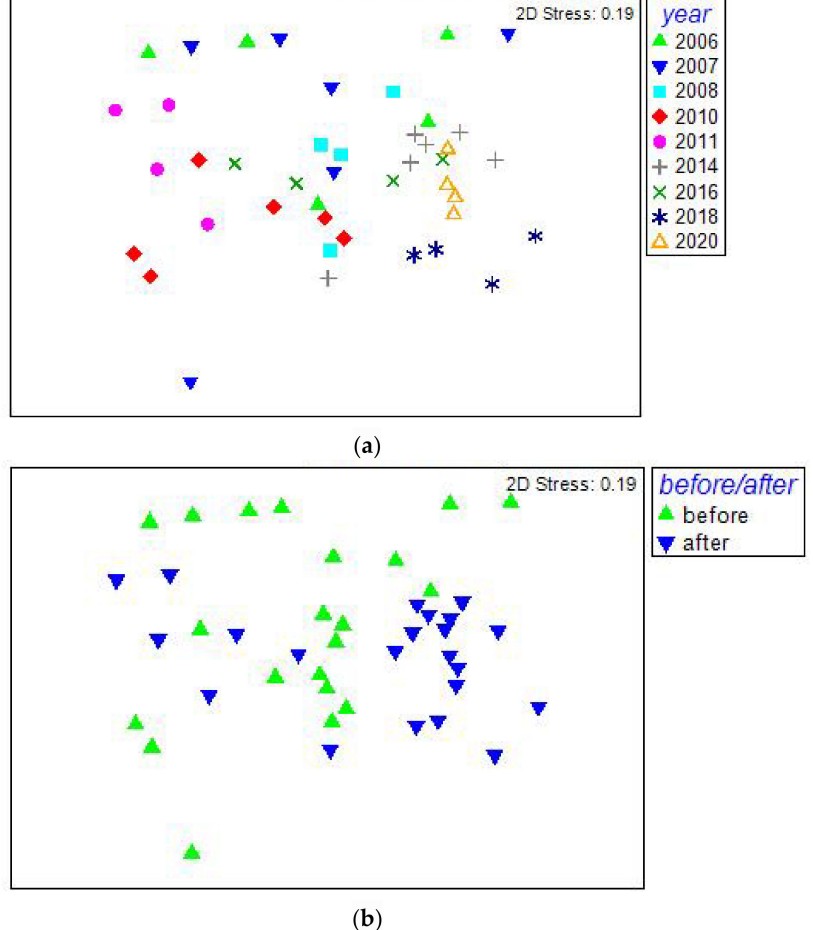

(**a**)

(**b**)

**Figure 3.** *Cont.*

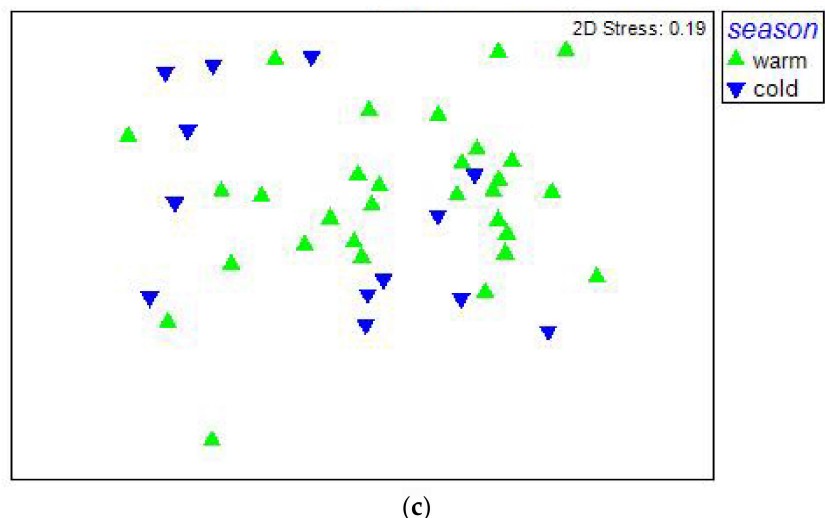

(**c**)

**Figure 3.** Nonmetric dimensional scaling ordination plots of species' relative abundance in fish of the Feitsui Reservoir. (**a**) MDS indicated differences in the fish assemblage in 2008, 2011, and 2014, (**b**) MDS plot of the fish assemblage for the years before and after 2010 indicated a partial overlap between the two groups, (**c**) The cold and warm seasons contained no exclusive taxa, contributing to the dissimilarity between the groups.

## 4. Discussion

Habitat loss and degradation caused by dam construction affect native fish populations and assemblages [36–38]; reservoirs are dynamic systems, and river–reservoir interfaces may provide refuge habitats during stochastic events, such as turbid inflows [39], in which fish use turbid water for cover and food resources [40,41]. This study uses multivariate statistical techniques to evaluate the spatial and temporal variations in the water quality of the Feitsui Reservoir. Based on the PCA results, we categorized years into two groups (cold and warm seasons) and assorted nine monitoring sites into three groups based on water quality characteristical. The temporal and spatial similarities and groupings determined in this study could promote the design of an optimal monitoring strategy that would allow for a lower monitoring frequency, several sampling stations, and related costs. While employment of PCA and PCs did not result in substantial data reduction, they enabled us to collect and identify factors and sources accountable for variations in water quality. Two variables obtained from the PCs indicated that the parameters accountable for water quality change were mainly related to algal count, Chl-a, TP, and Turbidity. The discriminant analysis provided favorable temporal and spatial results. Varifactor 1, which explained 64.5% of the variance, had strong positive loadings (>0.70) on algal count, a strong negative loading on SD, and a moderately negative loading on EC, which can be interpreted as resulting from mineral components on the surface water of the reservoirs. This variable indicates natural sources (inflows, soil weathering, and runoff) for the ionic groups in the reservoir.

We identified long-term change in the fish assemblage in the riverine zone (upper reach), transitional zone (midstream), and lacustrine zone (reach lower) environments in the Feitsui Reservoir. Generally, the effects of impoundment on fish assemblage were more pronounced in the lacustrine zone than in the lotic. The fish assemblages changed significantly over time within each habitat type. However, the degree of seasonal variation was similar; cold and warm season fish populations and assemblages demonstrated no significant difference. However, for *H. nobilis*, *C. cuvieri*, *S. macrops*, *C. carpio*, and *H. Leuciscus*, a significant difference ($p < 0.05$) was observed among their major habitats in the pre- and post 2010 years, and these differences persisted over time. *H. nobilis* is a large, deep-bodied cyprinid introduced from eastern Asia [42,43]. The fish primarily feed on zooplankton and large-sized phytoplankton with granule sizes of 17–3000 μm [44–47]. This advantage

confers great influences on the biomass and structure of the planktonic community, which, conversely, can trigger other trophic cascading effects on other planktivorous fish species and top predators. *C. cuvieri* prefers slow running water at lower and middle reaches, feeds on a wild range of food, including plants, diatom, crustacean, and aquatic insects. Nakamura (1969) [48] pointed out that the spawning migration of *C. cuvieri* was often limited to the waterside of the lake and that the aquatic plant area spread along the Feitsui Reservoir riverine zone (upper reach) plays an important role as the site. *S. macrops* can complete their life history without migration, so would seem not to be affected by dams [49]. *S. macrops* occurred downstream of the Shuaishui Stream of China [50]. On the other hand, *C. carpio*, a freshwater fish native to eastern Europe and central Asia, is one of the most invasive species in the world. The life-history of carp is one characterized by flexibility, with long breeding seasons (up to 9 months) and the ability to spawn multiple times each year [51–53]. The impacts of common carp include the destruction of aquatic vegetation, which decreases the diversity and abundance of invertebrates [54–56], increased water turbidity, and eutrophication [57,58]. Lastly, in China, *H. leuciSculus* occupies a wider range of freshwater habitats, such as rivers, lakes, reservoirs, and even pools [59,60]. The results of the cluster analysis indicated that the species that most significantly contributed to the fish assemblage from 2006 to 2020 were *H. nobilis*, *M. tilapia*, and *C. carpio*. The longitudinal patterns in diversity and the distribution of stream fish along upstream–downstream gradients and the changes in food sources along the river continuum constrain the trophic groups of aquatic organisms within communities. Fish species of generalized invertivores are expected in upstream areas, while omnivores, detritivores, herbivores, and piscivores become more abundant further downstream in a river basin [61–63].

In addition, nonnative species were found to play considerable roles in driving changes in the composition of fish. In the pre-2014 years, *P. managuensis* was essentially absent. However, in 2014–2020, *P. managuensis* appeared in June and the population increased rapidly to peak abundance in the fall. *P. managuensis* is piscivorous and indigenous to Honduras and Costa Rica, is a highly aggressive piscivore and an alien species for the Feitsui Reservoir. Introduced by sport fishing and aquaculture activities, they have already impacted other watersheds where they were introduced. This exotic species are likely to aggravate biodiversity loss of the Feitsui Reservoir. Agasen et al. (2006) [64] state that *P. managuensis* is as a predator that eats small fish and is very aggressive. The increase in the richness of nonnative fish species over time is a trend observed in many alter ecosystems [65–67]. We expected to identify a pronounced increase in nonnative fish species in the reservoir.

*P. managuensis* was first recorded in the Feitsui Reservoir in 2014. In the following years, the *C. cuvieri* population has been declining year on year. According to Effendie (2002) [68], for *P. managuensis*, the larger the size of the fish, the more varied the types of food; small fish tend to eat phytoplankton that is adjusted to the mouth opening, and the popular foods are Chlorophyta and Charophyta. However, in adulthood animals are also eaten. Larval fish compete with the larvae of other species, as well as juveniles and adults of other species when their diets overlap [69–73]. Considering that all riverine fishes feed on planktonic organisms during their early life stages [73,74], native fish larvae may be more vulnerable to the food web effects of invasive fish.

Taki (1978) [75] reported that in Southeast Asia, the distributional summit of cyprinids may contribute to 40% or more of the species in a watershed. In addition, Cyprinidae was the dominant family in Gangapur Dam [76], India. In Taiwan, Cyprinidae was also the dominant family, with 38 fish species belonging to 8 orders and 12 families collected in the Feitsui Reservoir, the majority belonging to Cypriniformes and Cyprinidae (accounting for 21 species). Overall, Cyprinidae account for over 71% of the total fish biomass and almost 72% of the pelagic fish biomass in the Feitsui Reservoir.

Because the bias was identical for all sampling years, our results indicate that the 14-year scale of this study was sufficient to include at least one turnover of individuals for nearly all specific individuals. The findings regarding the contributions of biotic and

abiotic factors in structuring fish assemblages have meaningful implications with respect to changes in the overall composition of the aquatic community and link the influence of these factors to the fish assemblage functional structures in reservoirs. Therefore, we highlight the need for studies of the effects of human activities on habitats and the available structures of fish assemblages to guide reservoir management and conservation projects.

## 5. Conclusions

The assemblages could be categorized into two groups (cold and warm seasons) through the principal component analysis and a nonmetric multidimensional scaling. Three parameters, SD (R = −0.04), Chl-a (R = 0.01), and algal count (R = 0.19), were found significantly correlated with the season. A higher algal count was observed in the warm season than in the cold season. The longitudinal and vertical gradients were found for the attributes and the structure of the fish assemblage of the Feitsui Reservoir as expected. There were four major findings related to the fish accumulation during the research period: (a) *S. macrops* had the highest population variation for the years before and after 2010 (>10%; SIMPER); (b) *O. mossambicus* was dominant in assemblage structures after 2010; (c) *P. managuensis* was first recorded in the Feitsui Reservoir in 2014; and (d) in the following years, the *C. cuvieri* population was declining year by year. The latter two findings reflects the importance of long-term fish monitoring for investigating the effect of nonnative fish species on native fish species composition.

**Author Contributions:** C.Y.-R. carried out the examine changes in fish assemblage structure in Feitsui Reservoir over 14-years post-impoundment period using a sentinel-site approach and performed the PDA statistical analysis. H.W.-S. conceived the study and participated in the experiment and coordination. H.C.-K. helped proofread the manuscript. C.C.-Y. conceived the study and helped proofread the manuscript. All authors have read and agreed to the published version of the manuscript.

**Funding:** This research was sponsored by Taipei Feitsui Reservoir Administration and National Taiwan University (NTU), Taipei, Taiwan.

**Institutional Review Board Statement:** This article does not contain any studies with animals performed by any of the authors.

**Data Availability Statement:** The data presented in this study are available in article material.

**Acknowledgments:** The authors wish to thank Taipei Feitsui Reservoir Administration and National Taiwan University (NTU) for their kindly support.

**Conflicts of Interest:** The authors declare that they have no conflict of interest.

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
