# Peer review of "Spatial and Temporal Variations in Fish Assemblage in Feitsui Reservoir, in Northern Taiwan, from 2006–2020"

_water, doi:10.3390/w14030498_

Round 1

Reviewer 1 Report

The paper presented for review, entitled "Spatial and temporal variations in fish assemblage in a reservoir: A long-term study" is a typical monitoring work. It takes into account both the seasonal and long-term variability of hydrochemical conditions as well as the variability in populations of individual fish species over a period of 14 years. The research was conducted on the Feitsui Reservoir which is the second-largest reservoir in Taiwan with a surface area of 303 km2. During the 14-year study, the authors maintained the same nine test stands. Unfortunately, there were no nitrogen determinations among the physicochemical parameters determined, and it is a pity because the obtained results indicate that it is a eutrophied reservoir, and nitrogen is one of the biogenic elements. Many years of monitoring of physicochemical parameters make it possible to determine the rate of eutrophication, but only taking into account all the necessary parameters, and in my opinion nitrogen is also one of them. The high concentration of total phosphorus (mean TP of 13.20 ± 7.52 mg L−1) is also interesting; it would be worth to provide the method used for determining the concentrations.

The biological studies related to the variability in the population of individual fish species are undoubtedly interesting and show how the species composition of fish in the reservoir changes over the course of 14 years. The authors identified 38 species of fish. The fish assemblage was dominated by species belonging to Cyprinidae (71%). The authors also referred to nonnative species that appear in such tanks.

To sum up, monitoring studies are very important to understand the functioning of ecosystems and therefore should be carried out, but the longer such studies, the more information can be obtained. I believe that the work submitted for review should be published.

Author Response

Dear reviewer,

Thank you for your review, we have revised and expanded the paper to respond to your questions. The questions have been answered as requested. Thank you so much for your comment.

best regards

Reviewer 2 Report

  • The abstract requires editing. My suggestion:

Reservoirs are artificial ecosystems which modify the hydrological and environmental components nearby. Long-term monitoring of fish assemblages in reservoirs may provide key information to understand how these artificial ecosystems allow fish population fluctuations and identify proper conservation strategies. A sentinel-site approach method assessed changes in fish assemblages in the Feitsui Reservoir (1,024 ha) over 14 years including the periods 2006-2008, 2010-2011, 2016, 2018, and 2020. Fish assemblages, sampled with gill nets, were composed by 38 species (families and orders?? numbers) with Cyprinidae as dominant family. A principal component analysis (BUT, consider CCA) and a nonmetric multidimensional scaling categorized the assemblages into two groups (cold and warm seasons), the Bray-Curtis similarity value was 70%......Up to here the abstract requires to include key information. When referring to Tilapia, is necessary to include the scientific name. What do authors imply with “The findings of this and previous research indicate that the fish assemblage in the Feitsu Reservoir is relatively stable”? Which are the indicators of stability?

  • I suggest the following title: Spatial and temporal variations in fish assemblage in the Feitsui reservoir, in northern Taiwan, from 2010-2020

  • The last paragraph of the Introduction must only have the objectives. Something like this: The aim of this study was to examine changes in fish assemblage structure in the Feitsui Reservoir, Taiwan, for 14 years post impoundment, using a sentinel‐site approach. We analyzed differences in fish assemblages throughout years to identify whether long-term variation associated with environmental conditions were evident in the fish populations by (1) quantifying the extent and dynamics of spatial changes, and changes in the correlation between water quality and fish, and by (2) identifying changes in fish assemblage structures due to the influence of exotic fish species.

  • In Materials and methods. Please, clarify if the nets used were all the very same for the whole sampling. If other types of nets were used, you need to clarify that because different nets and locations may catch different types of fish. Also, in line 120, you mentioned that fish were identified to species, weighed and measured. But, here, it is very important to mention which taxonomic keys were used to identify fish. You need to quote taxonomic fish authorities. Also, you need to mention how fish were measured and weighed, by using tape or a table for length (which length, total, standard, furcal?), which weight (total, gutted?).
  • Why using PCA instead of using CCA? This latter (CCA) is better when attempting to relate the presence of given species and the environmental variables at a given place.
  • The Discussion section is surprisingly too short for the coverage of the work. Why not to compare the results of this study with results from others in other geographic regions?
  • What is the considerable roll nonnative species play in driving changes in the fish assemblages’ composition? For instance, P. managuensis appeared before 2014 but how this species affected the native species? And how this very same species has affected other reservoirs in Taiwan or in any other place in the world?
  • I strongly recommend authors to discuss further the implications of their finding and compare with results of other reservoirs in Taiwan or other geographic regions.
  • Discuss why Cyprinidae is dominant in these environments and which is the role of this family in relation to the other fish families in the reservoirs.
  • Given the magnitude of the work, I suggest a Conclusions section. But, please, only include the key findings without any discussion nor any quotation in this section. Thanks.

Author Response

(The authors gave the same response as above.)

Round 2

Reviewer 2 Report

Authors addressed most of my suggestions. However, there are some that are not addressed yet:

*Authors need to mention how fish were measured (mm, cm?) Mention what total length is.

*It is necessary to mention that taxonomic key were used to identify fish.

*Authors used PCA. However, PCA IS NOT A STATISTICAL method but a MULTIVARIATE method. I suggested to use Canonical Correspondence Analysis (CCA) if authors are dealing with enviromental variables and dimensional space of fish (species, abundance, etc). However, authors did not respond why not using CCA instead of PCA.

Author Response

Dear reviewer,

Thank you for your comments, we have revised and expanded the paper to respond your questions.

Sincerely,

Round 3

Reviewer 2 Report

Authors followed my suggestions accordingly. I have no further suggestions other than just remove the very first sentence of the Conclusions section ("Based on the research results in Feitsui Reservoir, it can be concluded that:"). Just begin with the first paragraph and continue to the second one and please, remove the numbers.

Author Response

(The authors gave the same response as above.)
